# COVID-19 Vaccine Safety in Cancer Patients: A Single Centre Experience

**DOI:** 10.3390/cancers13143573

**Published:** 2021-07-16

**Authors:** Alfred Chung Pui So, Harriet McGrath, Jonathan Ting, Krishnie Srikandarajah, Styliani Germanou, Charlotte Moss, Beth Russell, Maria Monroy-Iglesias, Saoirse Dolly, Sheeba Irshad, Mieke Van Hemelrijck, Deborah Enting

**Affiliations:** 1Department of Oncology, Guy’s and St Thomas’ NHS Foundation Trust, London SE1 9RT, UK; harriet.mcgrath@gstt.nhs.uk (H.M.); jting@gstt.nhs.uk (J.T.); krishnie.srikandarajah@gstt.nhs.uk (K.S.); styliani.germanou@gstt.nhs.uk (S.G.); saoirse.dolly@gstt.nhs.uk (S.D.); sheeba.irshad@gstt.nhs.uk (S.I.); 2Translational Oncology & Urology Research (TOUR), School of Cancer and Pharmaceutical Sciences, King’s College London, London SE1 9RT, UK; charlotte.moss@gstt.nhs.uk (C.M.); beth.russell@gstt.nhs.uk (B.R.); maria.j.monroy_iglesias@kcl.ac.uk (M.M.-I.); mieke.vanhemelrijck@gstt.nhs.uk (M.V.H.)

**Keywords:** COVID-19 vaccine, cancer, side effects, reactogenicity

## Abstract

**Simple Summary:**

Although COVID-19 vaccine side effects are generally well tolerated, information on cancer patients is lacking due to their exclusion from original clinical trials. The aim of our study was to report on the safety of COVID-19 vaccines in our cancer patients. Data on vaccine side effects from our London cancer center was collected from 8 December 2020 to 28 February 2021. Reassuringly, we observed that cancer patients tolerated the first dose of COVID-19 vaccine very well with minimal serious side effects. Similar to the vaccine clinical trials, the most common side effects were having a sore arm, tiredness, and headaches.

**Abstract:**

Emergency approval of vaccines against COVID-19 provides an opportunity for us to return to pre-pandemic oncology care. However, safety data in cancer patients is lacking due to their exclusion from most phase III trials. We included all patients aged less than 65 years who received a COVID-19 vaccine from 8 December 2020 to 28 February 2021 at our London tertiary oncology centre. Solicited and unsolicited vaccine-related adverse events (VRAEs) were collected using telephone or face-to-face consultation. Within the study period, 373 patients received their first dose of vaccine: Pfizer/BioNTech (75.1%), Oxford/AstraZeneca (23.6%), Moderna (0.3%), and unknown (1.1%). Median follow-up was 25 days (5–85). Median age was 56 years (19–65). Of the patients, 94.9% had a solid malignancy and 76.7% were stage 3–4. The most common cancers were breast (34.0%), lung (13.4%), colorectal (10.2%), and gynaecological (10.2%). Of the patients, 88.5% were receiving anti-cancer treatment (36.2% parenteral chemotherapy and 15.3% immunotherapy), 76.1% developed any grade VRAE of which 2.1% were grade 3. No grade 4/5 or anaphylaxis were observed. The most common VRAEs within 7 days post-vaccination were sore arm (61.7%), fatigue (18.2%), and headaches (12.1%). Most common grade 3 VRAE was fatigue (1.1%). Our results demonstrate that COVID-19 vaccines in oncology patients have mild reactogenicity.

## 1. Introduction

The coronavirus disease 2019 (COVID-19) pandemic, caused by the severe acute respiratory syndrome coronavirus 2 (SARS-CoV-2), continues to impact cancer care and its patients worldwide. While many patients have asymptomatic or mild infections, some develop severe COVID-19 pneumonitis and its related complications. Pooled data analysis from observational studies have shown increased mortality from COVID-19 in oncology patients, particularly those with haematological malignancies [1]. The relationship between COVID-19 and cancer is complex with multiple confounding risk factors such as age and certain comorbidities (e.g., obesity, chronic lung disease) [2,3]. Aside from infection-related outcomes, the impact of the COVID-19 pandemic on cancer care is also widespread. This includes challenges with reduced services, contraction in routine screening programs, staff shortages, reduced health-seeking behaviours, and interruptions in cancer-related treatments [4]. These potential delays in early cancer diagnosis and management puts patients at risk of presenting with more aggressive disease, resulting in an excess of avoidable cancer-related deaths [5,6].

The advent of COVID-19 vaccines provides an opportunity to return to pre-pandemic oncology care. In the UK, there are currently three vaccines available: Pfizer/BioNTech (BNT162b2), Oxford/AstraZeneca (ChAdOx1 nCoV-19), and Moderna (mRNA-1273). Unfortunately, oncology patients have been under-represented in phase 3 trials due to the exclusion of patients with a history of active cancer, those who are immunosuppressed (including cytotoxic agents, long-term steroids), or recently received immunoglobulins and/or blood products [7,8,9]. Therefore, the data regarding the safety and efficacy of COVID-19 vaccines in cancer patients is lacking. Currently, the safety data in cancer patients is limited to a few small studies with no new safety signals observed [10,11]. Despite the uncertainty, multiple oncology groups have provided recommendations favouring the uptake of these vaccines due to the life-threatening risk of severe COVID-19 [12,13,14,15,16]. It was also recently shown that patients with cancer in the UK have been more severely impacted by COVID-19 with higher mortality rates compared to other European countries [17]. This likely reflects the increased frailty of UK cancer patients and highlights their need for vaccine prioritisation. Here, we report the initial results on the safety profile of COVID-19 vaccines in oncology patients under 65 years old. We recently reported the safety profile following Pfizer/BioNTech vaccine in older patients in the SOAP-02 study [10].

## 2. Materials and Methods

This was a retrospective cohort study of patients with cancer at a tertiary cancer centre in London, UK. Between 8 December 2020 and 28 February 2021, we collected outcome data for Guy’s Cancer patients aged 65 years or under who received at least one dose of COVID-19 vaccine. Exclusion criteria included patients with missing data on adverse events and age <18 and >65 years. Potentially eligible patients were extracted from our electronic database using a clinician assessment form for when they received the vaccine and were seen at the cancer centre. Patient demographics, oncological characteristics, and treatment information were extracted using our local oncology records. We defined current or recent treatment as within 28 days of receiving the vaccine, with the exception of immunotherapy, which is within 6 months. Prior infection status from COVID-19 was also reported and categorised into confirmed and suspected based on nasopharyngeal PCR or serological test and typical symptoms of COVID-19 infection. Data collection was approved under Guy’s Cancer Cohort (Reference number: 18/NW/0297) [18].

Both solicited and unsolicited adverse events within and after 7 days of vaccination were collected from face-to-face or telephone consultations. Solicited adverse events included pain at injection site, sore arm, local erythema, fever, fatigue, headaches, chills, arthralgia, myalgia, diarrhoea, nausea, vomiting, flu-like symptoms, and lymphadenopathy. Unsolicited adverse events were recorded when a patient spontaneously volunteered symptoms during consultation that had not been categorised as a solicited event or when a clinician identified signs and symptoms that may be related to the COVID-19 vaccines. Adverse events were graded according to Common Terminology Criteria for Adverse Events (CTCAE) v5.0.

Descriptive statistics were used for patient demographics, oncological characteristics, and vaccine safety outcomes. Multinomial logistic regression analysis was used for the following independent variables: age ≥55, gender, BMI ≥30, prior COVID-19 infection status, metastatic cancer, type of vaccine received, and whether they were having active systemic anti-cancer therapy or not. Minimal necessary adjustments on covariates were identified using directed acyclic graphs (DAG) on DAGitty (http://www.dagitty.net/dags.html access on 17 May 2021) (Table A1 and Figure A1 in Appendix A). Due to our small sample size and concerns of false positives from multiple comparisons, we used the Bonferroni correction to adjust the alpha cut-off for statistical significance at 0.005. Statistical analysis was performed on SPSS. No statistical analyses were performed on specific tumour groups and comorbidities due to the heterogeneous population.

## 3. Results

### 3.1. Patients

Between 8 December 2020 and 28 February 2021, 373 patients were included for data analysis with a median follow-up of 25 days (range 5–85 days). Of the 373 patients, only four had received a second dose of vaccine. Distribution of vaccine types were Pfizer/BioNTech 75.1% (*n* = 280), Oxford/AstraZeneca 23.6% (*n* = 88), Moderna 0.3% (*n* = 1), and unknown 1.1% (*n* = 4). Of the total cohort, 7.8% and 7.0% had confirmed and suspected previous COVID-19 infection respectively. At the time of data cut-off, two patients had positive COVID-19 PCR tests 1- and 2-days post-vaccination, respectively. One patient remained asymptomatic and another was admitted to hospital with severe COVID-19 pneumonitis. Median age was 56 years (range 19–65 years), 62.5% were female, 67.8% were Caucasian, mean BMI was 28 (range 15.1–64.6), and median indices of multiple deprivation index decile was 5 (range 1–10) (Table 1).

Of all patients, 94.9% had a solid malignancy, of which 76.7% were stage 3–4. Breast cancer was the most prevalent diagnosis (34.0%), followed by lung (13.4%), colorectal (10.2%), and gynaecological cancers (10.2%) (Table 2). The most common haematological malignancy was lymphoma (2.4%). Of all patients, 88.5% were receiving anti-cancer treatment during vaccination (36.2% parenteral chemotherapy, 23.6% hormone therapy, and 15.3% immunotherapy). Of the 15.3% patients receiving immunotherapy, six were on a combination of anti-PD-1/CTLA-4 regimens. Five patients with haematological malignancies were on B-cell depleting therapies (i.e., rituximab, obinutuzumab). At time of vaccination, 61.9% of patients were receiving systemic therapy with palliative intent and 11.5% had no active treatment. The average timing in the delivery of any systemic anti-cancer therapy prior to and after vaccination was 10 and 6 days respectively. With regards to patient comorbidities, 9.7% have diabetes, 5.6% autoimmune conditions, and 1.1% HIV (Table 3). Of all patients, 7.8% had an additional cancer diagnosis that was either currently in remission, surveillance, or not actively treated in comparison to their primary cancer.

### 3.2. Vaccine-Related Adverse Events

As only four patients received their second vaccine dose at data cutoff, we will only report the safety results following the first dose. In total, 76.1% patients developed any grade adverse events, of which 2.1% had grade 3 events and 1.6% experienced symptoms after more than 7 days post-vaccination (Table 4, Figure 1 and Figure 2). No grade 4–5 events or anaphylaxis were observed. The incidence of total any grade local and systemic adverse events were 61.7% and 33.8% respectively. The most common any grade adverse events within 7 days post-vaccination were pain at injection site or sore arm (61.7%), fatigue (18.2%), headaches (12.1%), myalgia (8.3%), and fever (5.6%). Median duration of solicited any grade local and systemic adverse events were 2 days (range 1–28). The most common grade 3 adverse event was fatigue (1.1%). Of the eight patients who experienced grade 3 adverse events, four had breast cancer, five had stage 3–4 disease, and three had prior COVID-19 infection (*n* = 2 suspected; *n* = 1 confirmed). The range of current systemic therapy of those patients included chemotherapy (*n* = 3), immunotherapy (*n* = 1), hormone therapy (*n* = 1), PARP inhibitor (*n* = 1), anti-EGFR tyrosine kinase inhibitor (*n* = 1), and no active treatment (*n* = 0). The frequency of unsolicited AEs was 8.8% with the most common symptom being dyspnoea (1.1%). Of interest, three patients receiving immunotherapy experienced new adverse events, which included worsening of pre-existing grade 1 pruritus, grade 2 transaminitis, and grade 2 hypocortisolism.

Four grade 3 adverse events occurred that required hospital admission. One patient with breast cancer on hormone therapy developed grade 3 transaminitis with liver capsule pain from suspected reactivation of hepatitis B. One patient developed grade 3 diarrhoea and urosepsis 12 h post-vaccination. One patient with locally advanced breast cancer developed febrile neutropaenia on day 9 post-chemotherapy, the fever of which was likely vaccine-related with no evidence of infection. Finally, one patient with metastatic bladder cancer developed recurrent pulmonary embolism 2 weeks post-vaccination (Pfizer/BioNTech), likely related to multiple risk factors including dehydration and reduced mobility from grade 3 flu-like symptoms, and fatigue and interruption in anticoagulation therapy due to haematuria. There was no incidence of vaccine-induced immune thrombotic thrombocytopaenia in our total cohort.

Of the total cohort, 349 patients were included for further analysis with respect to individual risk factors and vaccine reactogenicity (Table 5, Table 6 and Table 7). We excluded patients with haematological malignancies (*n* = 19) and incomplete datasets (*n* = 6, missing data for BMI). Individuals were analysed for their risk of developing total any grade vaccine-related adverse events, grade ≥2 adverse events, and any grade systemic adverse events. Male patients were less likely to report any vaccine-related adverse events compared to females (OR 0.426 [95%CI 0.259–0.699]; *p* = <0.001). Patients receiving immunotherapy within 6 months of vaccination appear to be at a lower risk of developing any vaccine-related adverse events as well (OR 0.495 [95%CI 0.256–0.958]; *p* = 0.0037). Age ≥55, BMI ≥30, presence of 1 or more comorbidities, prior COVID-19 infection, having metastatic cancer, receiving chemotherapy during time of vaccination, and receiving the Pfizer/BioNTech vaccine (compared to Oxford/AstraZeneca) did not influence risk of developing any grade adverse events. With regards to developing grade ≥2 adverse events, older patients (≥55 years old) (OR 0.481 [95%CI 0.237–0.974]; *p* = 0.042) and those receiving the Pfizer/BioNTech vaccine were at a lower risk but not statistically significant (OR 0.366 [95%CI 0.177–0.758]; *p* = 0.007). There was also a general trend towards lower incidence of grade ≥2 adverse events in patients with metastatic cancer (OR 0.493 [95%CI 0.238–1.021]; *p* = 0.057). Negative independent predictors of developing vaccine-related systemic adverse events include being male (OR 0.632 [95%CI 0.400–0.999]; *p* = 0.049), having metastatic cancer (OR 0.548 [95%CI 0.347–0.867]; *p* = 0.010), receiving chemotherapy within 28 days of vaccination (OR 0.373 [95%CI 0.221–0.629]; *p* < 0.001), and receiving the Pfizer/BioNTech vaccine (OR 0.452 [95%CI 0.274–0.747]; *p* = 0.002). There was no association between prior COVID-19 infection (both suspected and confirmed) and risk of vaccine-related adverse outcomes.

## 4. Discussion

These initial results from our study support the favourable safety profile that was also observed during phases 1–3 testing of COVID-19 vaccines, with reactogenicity generally mild or moderate [7,8,19,20,21]. This has been similarly reported in recent observational studies on the safety profile of the Pfizer/BioNTech vaccine in cancer patients [10,11]. In the SOAP-02 study, 46% of cancer patients (*n* = 75/140) had any grade adverse events following the first dose of the Pfizer/BioNTech vaccine with injection-site pain within 7 days being the most common (16.4%; *n* = 23/140) [10]. The study had proportionately more patients with haematological malignancies (*n* = 50/140) and were generally older with a median age of 73 (IQR 31.3–50.0) [10]. Interestingly, one patient previously on immunotherapy developed grade 4 transaminitis of unclear cause 3 weeks post-vaccination [10]. A recent study by a group in Israel observed no increased risk of serious adverse events in cancer patients treated with immunotherapy after receiving two doses of the Pfizer/BioNTech vaccine compared to healthy controls [11]. In their cohort of 134 patients, the most common side effect was pain at injection site (21%) following the first dose [11]. Incidence of systemic adverse events was generally low including fatigue (4%), headache (3%), myalgia (2%), and chills (1%) [11]. Comparatively, the study had proportionately more patients with lung cancer (49.2%) and were older with a median age of 72 (range 29–93) [11]. In our cohort, we reported higher incidence of total and severe adverse events compared to both studies, which likely reflects the younger age of our patients (median age 56). These differences in adverse events between younger and older healthy adults have been previously reported in phase 1–3 clinical trials [7,8,18,19]. With the Pfizer/BioNTech vaccine, younger recipients (<55 years old) reported higher rates of local (88.7% vs. 79.7%) and systemic reactogenicity (82.8% vs. 70.6%) compared to older recipients (≥55 years old) following the first dose of vaccine [22]. This was also observed following the second dose but was generally less frequent. Similar trends were observed with the Oxford/AstraZeneca studies, with decreasing incidence of local and systemic symptoms from the 18–55 years age group (88% and 86%) to the 56–69 years group (73% and 77%) and the ≥70 years group (61% and 65%) [20]. These differences observed in reactogenicity between younger and older cohorts may be explained by a higher degree of symptom tolerance in older people and age-related decline in immune responses. In our cohort, we observed similar correlations with severity, where older patients (≥55 years old) had lower incidence of grade ≥2 vaccine-related adverse events compared to younger patients (OR 0.481 [95%CI 0.237–0.974]; *p* = 0.042). Unfortunately, our study focused only on patients ≤65 years old and therefore cannot reliably interpret the correlation of age and reactogenicity due to underrepresentation of patients >65 years old.

Compared to the vaccine trials, we observed lower incidence of local and systemic vaccine-related adverse events in our current cohort of patients with cancer [7,8,19,20,21]. In the Pfizer/BioNTech studies, total incidence of any grade local and systemic adverse events within 7 days of vaccination were reported as 84.7% and 77.4% respectively [22]. In the Oxford/AstraZeneca studies, incidence of local and systemic adverse events following the first dose of vaccine were around 61–88% and 65–86% [20]. A large prospective UK study looking at self-reported vaccine side effects in the community also observed lower incidence of events compared to the clinical trials [23]. Side effects were logged and reported by participants using a COVID Symptom Study app. Total local side effects following the first dose of Pfizer/BioNTech or Oxford/AstraZeneca vaccine were 71.9% and 58.7% respectively [23]. Total systemic adverse events following the first dose of Pfizer/BioNTech or Oxford/AstraZeneca vaccine were much less reported at 13.5% and 33.7% respectively [23]. A similar study in the US using a smartphone-based system reported the incidence of total local and systemic events following the first dose of Pfizer/BioNTech vaccine as 70.0% and 50.0% respectively [24]. In both the clinical trials and post-marketing surveillance studies, fatigue and headache were the most commonly reported systemic adverse event [7,8,19,20,21,22,23,24]. Similar distribution of systemic symptoms is observed in cancer patients, with fatigue and headache being most commonly reported [10,11].

Interestingly, current post-marketing observational studies have all seen a lower incidence of vaccine-related adverse events compared to their respective clinical trials. Evidently, several confounders exist including selection bias; reporting bias; differences in methodology, frequency, and intensity of monitoring in clinical trials; and variations in patient characteristics. One of the main confounders that may explain these differences in adverse events between clinical trials and real-world evidence is the cohort age. The majority of the observational studies have generally older participants that reflect current vaccination prioritisation, thus having lower incidence of vaccine-related adverse events. Real-world patients prioritised for the vaccine also generally have more comorbidities, although there is yet to be any correlation identified with COVID-19 vaccine reactogenicity [23]. Other potential risk factors for reactogenicity include gender and type of vaccine [23]. Female recipients were more likely to report any adverse events following either the Pfizer/BioNTech (OR 1.89 [95%CI 1.85–1.94]; *p* < 0.0001) or Oxford/AstraZeneca vaccines (OR 1.82 [95%CI 1.79–1.85]; *p* < 0·0001) [23]. Systemic adverse events were more common in individuals who received the Oxford/AstraZeneca vaccine compared to the Pfizer/BioNTech vaccine (OR 3.33 [95%CI 3.29–3.37]; *p* < 0.0001) [23]. This association was reversed with local adverse events, where individuals receiving the Pfizer/BioNTech were more common to report symptoms compared to those receiving the Oxford/AstraZeneca vaccine (OR 0.72 [95%CI 0.71–0.73]; *p* < 0.0001) [23]. Initial results from the Com-COV study comparing heterologous to homologous prime-boost regimens with Pfizer/BioNTech and Oxford/AstraZeneca vaccines also showed similar differences following the first dose [25]. These trends were also reproduced in our study with a reduced incidence of any adverse events in male patients (OR 0.426 [95%CI 0.259–0.699]; *p* = < 0.001) and reduced incidence of any systemic adverse events in patients who received the Pfizer/BioNTech vaccine (OR 0.452 [95%CI 0.274–0.747]; *p* = 0.002).

Reasons for potential lower vaccine-related reactogenicity in cancer patients compared to non-immunocompromised individuals may include overlapping symptoms with chronic disease and anti-cancer therapies. Patients with cancer may also have higher tolerance to certain symptoms such as pain, fatigue, and myalgia as a result of their chronic illness and may therefore not report it. However, the current evidence remains unclear. In SOAP-02, compared to healthy controls, there were lower incidence of local (52% vs. 36%) and systemic (32% vs. 25%) symptoms following the Pfizer/BioNTech vaccine in cancer patients [10]. On the other hand, cancer patients on immune checkpoint inhibitors receiving the Pfizer/BioNTech vaccine experienced similar systemic symptoms compared to healthy controls, with the exception of myalgia, which was more common in cancer patients [11]. Furthermore, while not a direct comparison, the UK app-based surveillance study reported the frequency of systemic symptoms as low as 13.5% with the Pfizer/BioNTech vaccine and 33.7% with the Oxford/AstraZeneca vaccine in the general population [23]. It is important to recognise that the design of this study relied heavily on individual reporting and may therefore have missed some severe adverse events if they were too unwell to use the app [23].

Another possible reason for lower vaccine-related adverse events is that cancer patients have reduced immunogenicity to the vaccine as recently demonstrated in the SOAP-02 study [10]. Reduced vaccine immunogenicity is nothing new in oncology patients and has been well observed with seasonal influenza vaccines [26,27,28,29]. This is most likely multifactorial as a result of host immune dysregulation from the cancer, immunosuppressive or immune-modulating treatments, bone marrow suppression, concurrent comorbidities, and a generally older and frailer patient population. In SOAP-02, serological non-responders to the Pfizer/BioNTech vaccine appeared to be more common in those who were receiving chemotherapy within 15 days pre-vaccination [10]. Interestingly, our patients who were receiving chemotherapy within 28 days of vaccination reported significantly less systemic vaccine-related adverse events compared to others (OR 0.373 [95%CI 0.221–0.629]; *p* < 0.001). We may be able to extrapolate these findings to suggest that patients with potentially compromised immune systems have both reduced vaccine immunogenicity and reactogenicity. Unfortunately, due to the observational nature of our study, we do not have the antibody titers of our vaccinated patients to do further regression analysis. This may be an important component to include in future studies. However, the correlation between vaccine immunogenicity and reactogenicity has historically been quite unclear with non-COVID-19 vaccines [30,31]. Several studies have suggested that immunisation outcomes may be independent of the systemic inflammatory process that underlies vaccine side effects [30]. Regardless, chemotherapy-induced myelosuppression extends to both myeloid and lymphoid lineages and should therefore dampen responses involved in both vaccine immunogenicity and reactogenicity [32,33].

Although our cohort observed higher grade 3 adverse events compared to the clinical trials, these are likely multifactorial and cannot be attributed to the vaccine alone. Due to our small sample size and the low incidence of serious adverse events (grade ≥ 3), we were unable to utilise any regression analysis to assess for risk factors. Of the eight patients who developed grade 3 vaccine-related adverse events, four had breast cancer, five had stage 3–4 disease, and three had prior COVID-19 infection. While there is inadequate evidence to draw any conclusions between cancer-related outcomes and vaccine reactogenicity, there has been emerging evidence to suggest that prior COVID-19 infection increases the risk of vaccine side effects [34]. In our subset of patients with prior confirmed (*n* = 29) and suspected (*n* = 26) COVID-19 infection, incidence of any vaccine-related adverse events was 75.9% and 73.1%, respectively, which was similar to our general cohort. However, our study was underpowered to detect any significant differences.

Finally, it is important to highlight some limitations of this study. The method in identifying vaccinated patients requires clinicians to complete the COVID-19 vaccine assessment tool in our local chemotherapy database during any patient contact. Therefore, patients may be missed from the data extraction due to incomplete assessments and those who are on prolonged surveillance intervals. This likely explains the underrepresentation of haematological malignancies and early-stage cancers in our cohort. Another limitation to our study is that the majority of our patients have not yet received a second vaccine dose due to current vaccination strategies to increase population uptake of the first dose.

## 5. Conclusions

Our initial findings can provide reassurance to both clinicians and patients and encourage increased uptake of the COVID-19 vaccine. With the emergence of new rare COVID-19 vaccine-related safety signals, such as immune thrombocytopaenic thrombotic syndromes, further post-marketing surveillance is warranted [35,36]. Our safety monitoring study is ongoing and we will expand the cohort with longer follow-up, including data following the second dose.

## Figures and Tables

**Figure 1 cancers-13-03573-f001:**
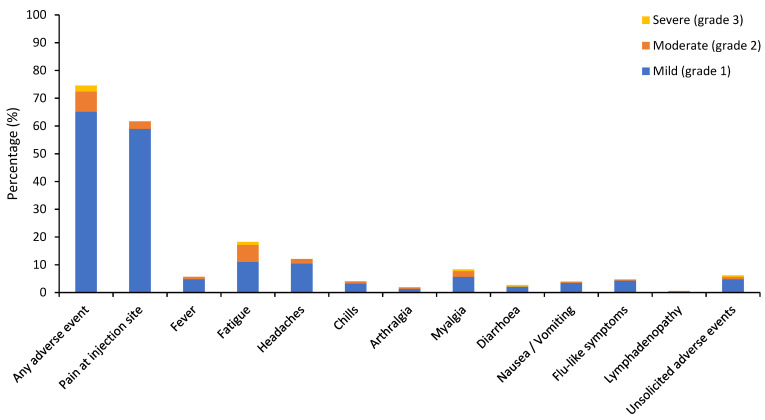
Vaccine-related adverse events within 7 days post-vaccination.

**Figure 2 cancers-13-03573-f002:**
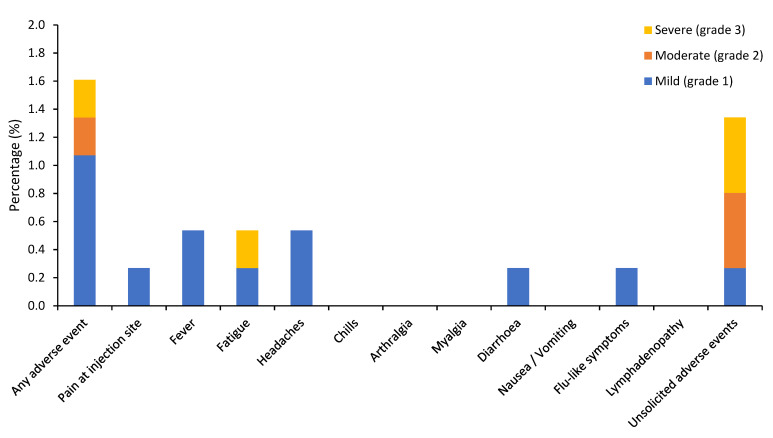
Vaccine-related adverse events after 7 days post-vaccination.

**Table 1 cancers-13-03573-t001:** Patient demographics.

Patient Demographics	Total (*n* = 373)
Sex—no. (%)	
Male	140 (37.5)
Female	233 (62.5)
Race or ethnic group—no. (%)	
White	253 (67.8)
Black	54 (14.5)
Asian	20 (5.4)
Arab	1 (0.3)
Iranian	2 (0.5)
Mixed	6 (1.6)
Unknown *	37 (10.2)
Median age—year. (range)	56 (19–65)
Age groups—no. (%)	
18–54	170 (45.6)
55–65	203 (54.4)
Mean BMI—no. (range)	28 (15.1–64.6)
BMI groups—no. (%)	
≤18.5 (underweight)	13 (3.5)
18.5–24.9 (normal)	112 (30.0)
25–29.9 (overweight)	138 (37.0)
30–34.9 (obese stage I)	57 (15.3)
35–39.9 (obese stage II)	29 (7.8)
≥40 (obese stage III)	18 (4.8)
Unknown	6 (1.6)
Median IMD decile—no. (range)	5 (1–10)

* No data or unable to confirm with patient.

**Table 2 cancers-13-03573-t002:** Patient comorbidities.

Patient Comorbidities	Total (*n* = 373)
Prior COVID-19 infection—no. (%) *	55 (14.7)
Yes	29 (7.8)
Suspected	26 (7.0)
Respiratory disorders—no. (%)	57 (15.3)
Asthma	33 (8.9)
COPD	20 (5.4)
Diabetes—no. (%)	36 (9.7)
Insulin-dependent diabetes	14 (3.8)
Non-insulin dependent diabetes	22 (5.9)
Cardio/Cerebrovascular disease—no. (%)	110 (29.5)
Hypertension	95 (25.5)
Ischaemic heart disease	12 (3.2)
Stroke	5 (1.3)
Autoimmune conditions—no. (%)	21 (5.63)
Inflammatory bowel disease	6 (1.6)
Systemic lupus erythematosus	3 (0.8)
Rheumatoid arthritis	3 (0.8)
Others †	10 (3.2)
Chronic viral infections—no. (%)	6 (1.6)
HIV	4 (1.1)
Hepatitis C	0
Hepatitis B	2 (0.5)
Other cancers—no. (%) ‡	29 (7.8)
Breast	10 (2.7)
Prostate	4 (1.1)
Haematological	4 (1.1)
Colorectal	2 (0.5)
Head & Neck	2 (0.5)
Urological	2 (0.5)
Melanoma	1 (0.3)
Endocrine	1 (0.3)
Gynaecological	1 (0.3)

* Previous COVID-19 diagnosis defined as a positive COVID-19 PCR swab. Suspected COVID-19 diagnosis defined by classical symptoms without a positive PCR swab (mainly due to lack of routine swabbing during the first wave of the pandemic). † Other autoimmune conditions within our cohort: Sarcoidosis (*n* = 2), Multiple sclerosis (*n* = 2), Hashimoto’s thyroiditis (*n* = 1), Grave’s disease (*n* = 1), Myasthenia gravis (*n* = 1), Immune-thrombocytopaenic purpura (*n* = 1), Anti-phospholipid syndrome (*n* = 1), Autoimmune hepatitis (*n* = 1), Primary sclerosing cholangitis (*n* = 1), Lichen planus (*n* = 1), Fibrosing alopecia (*n* = 1), and Raynaud’s syndrome (*n* = 1). ‡ These are cancers that are either currently in remission, surveillance, or not actively treated in comparison to the primary cancer diagnosis.

**Table 3 cancers-13-03573-t003:** Oncological characteristics. SACT, systemic anti-cancer therapy. TKI, tyrosine kinase inhibitor.

Oncological Characteristics	Total (*n* = 373)
Tumour type (solid)—no. (%)	354 (94.9)
Breast	127 (34.0)
Lung	50 (13.4)
Gynae	38 (10.2)
Colorectal	38 (10.2)
Urological	34 (9.1)
Prostate	23 (6.2)
Melanoma	20 (5.4)
Others *	24 (6.4)
Tumour type (haematological)—no. (%)	19 (5.1)
Myeloma	8 (2.1)
Lymphoma	9 (2.4)
Leukaemia	2 (0.5)
Cancer stage for solid tumours—no. (%)	
1	15 (4.0)
2	53 (14.2)
3	69 (18.5)
4	217 (58.2)
Current treatment intent—no. (%)	
Radical	101 (27.1)
Palliative/Surveillance	231 (61.9)/31 (8.3)
Unknown	10 (2.7)
Current treatment regimen—no. (%) †	
Parenteral SACT	
Chemotherapy only Chemotherapy + Immunotherapy Chemotherapy + Target therapy Chemotherapy + Hormone therapy Chemotherapy + Target + Hormone Target therapy (anti-EGFR) Target therapy (anti-HER-2) Target therapy (anti-VEGF) Chemo-radiotherapy	90 (24.1) 11 (2.9) 25 (6.7) 5 (1.3) 3 (0.8) 3 (0.8) 30 (8.0) 4 (1.1) 1 (0.3)
Oral SACT (continuous)	
Chemotherapy Mtuli-target TKIs (inc. anti-VEGF) PARP inhibitors CKD4/6 inhibitors ALK inhibitors Anti-EGFR (T790M) BRAF/MEK inhibitors mTOR inhibitors RET inhibitors ATR inhibitors	21 (5.6) 12 (3.2) 9 (2.4) 25 (6.7) 3 (0.8) 2 (0.5) 3 (0.8) 2 (0.5) 2 (0.5) 1 (0.3)
Hormone therapy (total)	88 (23.6)
Immunotherapy	
Anti-PD-1 or Anti-PD-L1 Combination Anti-PD-1/CTLA-4	51 (13.7) 6 (1.6)
Haematological SACT	17 (4.6) ‡
No active treatment	43 (11.5)

* Other primary cancer diagnoses within our cohort: Hepato-pancreato-biliary (*n* = 6), Gastro-oesophageal (*n* = 4), Neuroendocrine tumours (*n* = 3), Mesothelioma (*n* = 2), Head and neck (*n* = 2), Endocrine (*n* = 2), Thymic (*n* = 2), Central nervous system (*n* = 1), Gastrointestinal stromal tumours (*n* = 1), and Appendiceal (*n* = 1). † Defined as SACT within ±28 days of vaccine (with the exception of immunotherapy, which is defined as within ±6 months of vaccine). ‡ *n* = 5 patients were on B-cell depleting therapies (i.e rituximab, obinutuzumab).

**Table 4 cancers-13-03573-t004:** Local and systemic adverse events reported after the first dose of the COVID-19 vaccine.

Total (*n* = 373)	within 7 Days Post-Vaccination—No. (%)				after 7 Days Post-Vaccination—No. (%)			
	Any Grade	Grade 1	Grade 2	Grade 3	Any Grade	Grade 1	Grade 2	Grade 3
Any adverse event	278 (74.5)	243 (65.1)	27 (7.2)	8 (2.1)	6 (1.6)	4 (1.1)	1 (0.3)	1 (0.3)
Local adverse events								
Pain at injection site/ Sore arm	230 (61.7)	220 (59.0)	10 (2.7)	0	1 (0.3)	1 (0.3)	0	0
Erythema	1 (0.3)	1 (0.3)	0	0	0	0	0	0
Systemic adverse events								
Fever *	21 (5.6)	18 (4.8)	3 (0.8)	0	2 (0.5)	2 (0.5)	0	0
Fatigue	68 (18.2)	41 (11.0)	23 (6.2)	4 (1.1)	2 (0.5)	1 (0.3)	0	1 (0.3)
Headaches	45 (12.1)	39 (10.5)	6 (1.6)	0	2 (0.5)	2 (0.5)	0	0
Chills	15 (4.0)	12 (3.2)	3 (0.8)	0	0	0	0	0
Arthralgia	7 (1.9)	5 (1.3)	2 (0.5)	0	0	0	0	0
Myalgia	31 (8.3)	21 (5.6)	8 (2.1)	2 (0.5)	0	0	0	0
Diarrhoea	10 (2.7)	8 (2.1)	0	2 (0.5)	1 (0.3)	1 (0.3)	0	0
Nausea/Vomiting	15 (4.0)	13 (3.5)	1 (0.3)	1 (0.3)	0	0	0	0
Flu-like symptoms	18 (4.8)	16 (4.3)	1 (0.3)	1 (0.3)	1 (0.3)	1 (0.3)	0	0
Lymphadenopathy	2 (0.5)	0	2 (0.5)	0	0	0	0	0
Other adverse events	28 (7.5)	18 (4.8)	3 (0.8)	2 (0.5)	5 (1.3)	1 (0.3)	2 (0.5)	2 (0.5)
Chest pain	1 (0.3)	1 (0.3)	0	0	0	0	0	0
Dyspnoea	4 (1.1)	4 (1.1)	0	0	0	0	0	0
GORD/Gastritis	2 (0.5)	1 (0.3)	1 (0.3)	0	0	0	0	0
Abdominal pain	3 (0.8)	3 (0.8)	0	0	0	0	0	0
Sore throat	3 (0.8)	3 (0.8)	0	0	0	0	0	0
Paraesthesia	2 (0.5)	2 (0.5)	0	0	0	0	0	0
Hot flushes	2 (0.5)	2 (0.5)	0	0	0	0	0	0
Hypotension	1 (0.3)	0	1 (0.3)	0	0	0	0	0
Tumour-pain†	2 (0.5)	2 (0.5)	0	0	0	0	0	0
Transaminitis	0	0	0	0	2 (0.5)	0	1 (0.3) ‡	1 (0.3) §
Urosepsis	1 (0.3)	0	0	1 (0.3) ¶	0	0	0	0
Anorexia	2 (0.5)	1 (0.3)	1 (0.3)	0	0	0	0	0
Febrile neutropaenia	1 (0.3)	0	0	1 (0.3) #	0	0	0	0
Cough	2 (0.5)	2 (0.5)	0	0	0	0	0	0
Dizziness	1 (0.3)	1 (0.3)	0	0	1 (0.3)	1 (0.3)	0	0
Euphoria	1 (0.3)	1 (0.3)	0	0	0	0	0	0
Weak arm	1 (0.3)	1 (0.3)	0	0	0	0	0	0
Hypocortisolism	0	0	0	0	1 (0.3)	0	1 (0.3) ‡	0
VTE	0	0	0	0	1 (0.3)	0	0	1 (0.3) **
Pruritus	1 (0.3)	1 (0.3) ‡	0	0	0	0	0	0

* Defined as subjective self-reported fever symptoms by patients. Those who did not have a recorded temperature either using a home thermometer or during clinical assessment were categorised as grade 1 fever. † Defined as the occurrence or worsening of pain at the location of a known malignant tumour. ‡ These adverse events occurred in patients (*n* = 3) who were receiving checkpoint inhibitors and were within 2 weeks of receiving the COVID-19 vaccine. These are known toxicities of immunotherapy and it is not known whether they are as a result of the vaccine increasing the incidence of immunotherapy-related side effects or not. In one patient (*n* = 1), his existing immunotherapy-related pruritus had transiently worsened over several days after receiving the vaccine and returned to normal without any intervention. § *n* = 1 patient developed grade 3 transaminitis and liver capsule that required hospital admission for symptom control and investigation. This patient was subsequently diagnosed with acute hepatitis B unrelated to current treatment or cancer diagnosis. Whether this is a result of a new diagnosis or reactivation of previous hepatitis B is unknown. ¶ *n* = 1 patient was admitted to hospital with grade 3 diarrhoea and urosepsis 12 h after receiving the COVID-19 vaccine. This is likely an incidental finding and the diarrhoea can be attributed to both urosepsis and the vaccine. # *n* = 1 patient developed febrile neutropaenia 1 day after receiving the COVID-19 vaccine and was admitted for management of suspected neutropaenic sepsis. She was on cycle 1, day 9 ddEC for stage 3 breast cancer. There was no source of infection identified and the fever was most likely a result of the vaccine. ** *n* = 1 patient was diagnosed with a recurrent pulmonary embolism (PE) around 2 weeks after receiving the COVID-19 vaccine. One week after receiving the vaccine, the patient developed grade 3 flu-like symptoms and fatigue, which resulted in reduced mobility and oral intake. He had a known diagnosis of metastatic bladder cancer and a previous PE for which he had been taking apixaban intermittently due to haematuria. The cause of the recurrent PE is likely multifactorial from previous PE, metastatic cancer, sub-therapeutic drug levels, reduced mobility, and dehydration.

**Table 5 cancers-13-03573-t005:** Risk of total any grade vaccine-related adverse events (*n* = 349). SACT, systemic anti-cancer therapy.

Risk Factors	Adjusted OR	95%CI (Lower)	95%CI (Upper)	*p*-Value
Age ≥55 (ref: age <55)	0.931	0.567	1.528	0.776
Male (ref: female)	0.426	0.259	0.699	<0.001 *
BMI ≥30 (ref: BMI <30)	0.935	0.535	1.632	0.812
Comorbidities (≥1) (ref: no comorbidities)	1.192	0.706	2.010	0.511
Prior COVID-19 infection (ref: no prior COVID-19 infection)	1.025	0.503	2.089	0.946
Metastatic cancer (ref: non-metastatic cancer)	0.848	0.493	1.458	0.551
Receiving active systemic anti-cancer therapy (ref: not receiving active SACT)	1.030	0.469	2.263	0.942
Receiving chemotherapy (within 28 days) (ref: not receiving chemotherapy within 28 days)	0.602	0.345	1.051	0.074
Receiving immunotherapy (within 6 months) (ref: not receiving immunotherapy within 6 months)	0.495	0.256	0.958	0.037
Pfizer vaccine (ref: receiving non-Pfizer/BioNTech vaccine)	0.929	0.522	1.652	0.801

* Statistically significant (alpha threshold of 0.005 after Bonferroni correction).

**Table 6 cancers-13-03573-t006:** Risk of any grade ≥2 vaccine-related adverse events (*n* = 349). SACT, systemic anti-cancer therapy.

Risk Factors	Adjusted OR	95%CI (Lower)	95%CI (Upper)	*p*-Value
Age ≥55 (ref: age <55)	0.481	0.237	0.974	0.042
Male (ref: female)	0.930	0.446	1.938	0.847
BMI ≥30 (ref: BMI <30)	0.797	0.346	1.835	0.594
Comorbidities (≥1) (ref: no comorbidities)	1.120	0.535	2.343	0.763
Prior COVID-19 infection (ref: no prior COVID-19 infection)	1.518	0.607	3.795	0.372
Metastatic cancer (ref: non-metastatic cancer)	0.493	0.238	1.021	0.057
Receiving active systemic anti-cancer therapy (ref: not receiving active SACT)	1.262	0.421	3.783	0.677
Receiving chemotherapy (within 28 days) (ref: not receiving chemotherapy within 28 days)	0.822	0.364	1.859	0.638
Receiving immunotherapy (within 6 months) (ref: not receiving immunotherapy within 6 months)	1.492	0.568	3.916	0.417
Pfizer vaccine (ref: receiving non-Pfizer/BioNTech vaccine)	0.366	0.177	0.758	0.007

* Statistically significant (alpha threshold of 0.005 after Bonferroni correction).

**Table 7 cancers-13-03573-t007:** Risk of any grade systemic vaccine-related adverse events (*n* = 349). SACT, systemic anti-cancer therapy.

Risk Factors	Adjusted OR	95%CI (Lower)	95%CI (Upper)	*p*-Value
Age ≥55 (ref: age <55)	0.803	0.521	1.240	0.323
Male (ref: female)	0.632	0.400	0.999	0.049
BMI ≥30 (ref: BMI <30)	1.065	0.655	1.733	0.799
Comorbidities (≥1) (ref: no comorbidities)	1.003	0.635	1.583	0.990
Prior COVID-19 infection (ref: no prior COVID-19 infection)	1.691	0.903	3.166	0.101
Metastatic cancer (ref: non-metastatic cancer)	0.548	0.347	0.867	0.010
Receiving active systemic anti-cancer therapy (ref: not receiving active SACT)	1.578	0.830	3.002	0.164
Receiving chemotherapy (within 28 days) (ref: not receiving chemotherapy within 28 days)	0.373	0.221	0.629	<0.001 *
Receiving immunotherapy (within 6 months) (ref: not receiving immunotherapy within 6 months)	0.662	0.345	1.270	0.215
Pfizer vaccine (ref: receiving non-Pfizer/BioNTech vaccine)	0.452	0.274	0.747	0.002 *

* Statistically significant (alpha threshold of 0.005 after Bonferroni correction).

## Data Availability

The data presented in this study are available on request from the corresponding author. The data are not publicly available due to ethical reasons.

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
