# Peer review of "COVID-19 Vaccine Safety in Cancer Patients: A Single Centre Experience"

_cancers, 2021, doi:10.3390/cancers13143573_

Round 1

Reviewer 1 Report

This paper provides a quite thorough review of previous studied of COVI-19 vaccines in cancer patients. 

Since so many previous studies have been published this paper can provide greater value if the side effects of the second dose are reported, especially since the side effects of the second dose are more severe and if included, will improve the significance of this paper.

The Discussion section is a bit long and should be shortened.

Author Response

Dear reviewer, thank you for your comments on our manuscript.

We understand that a major limitation of our paper is that it currently does not report the side effects of the second dose. This is because many of our younger patients have yet to receive the second dose due to our country’s (UK) vaccination strategy. Although some time has passed since our initial data collection, the data surrounding vaccine safety in oncological patients remains limited and we therefore want to report these initial results of the first dose in a peer reviewed journal to encourage confidence and address any vaccine hesitancy amongst the population. To address your comment, we have highlighted in the latest paragraph of the discussion that we do aim to collect data following the second dose with longer-term follow-up in the future: “Another limitation to our study is that the majority of our patients have not yet received a second vaccine dose due to current vaccination strategies to increase population uptake of the first dose.” (line 382-384).

As for the discussion section, we appreciate that it is long but that was done with the intention to have a comprehensive overview of the current understanding of vaccine safety in different groups of patients and predictors of adverse events. We would prefer the discussion to be kept at its current length but we are happy to shorten based on further suggestions from the editor. 

Reviewer 2 Report

The authors report on side effects after 1 COVID-19 vaccination in 373 cancer patients under age 65, most with a breast cancer. They share side effects and reactogenicity. The manuscript is well written but didn't take side effects after the second dose into account which is a limitation.

Author Response

Dear reviewer, thank you for your comments on our manuscript.

Please see our response to reviewer 1 – as we now have highlighted this limitation in our latest paragraph of the discussion: “Another limitation to our study is that the majority of our patients have not yet received a second vaccine dose due to current vaccination strategies to increase population uptake of the first dose.” (line 382-384). 

Reviewer 3 Report

Questions: - There appears to be a substantial amount of comorbidities. Were some of the effects noted following vaccination due to the comorbidities rather than the cancer? - Since 28 February until now, have any of the participants in this study, reported any side effects, ie. are side effects showing up 3-4 months later? - Human ethics information should be included - Are antibody titers known for the vaccinated individuals? Do symptoms correlate with antibody titer?

Author Response

Dear reviewer, thank you for your comments on our manuscript.

To address your first question, we have not observed any adverse events that were related to the comorbidities rather than the cancer. Majority of the adverse events reported were solicited and expected adverse events previously reported in the vaccine clinical trials. We have also done a logistic regression analysis to determine whether having one or more comorbidities was an independent risk factor for developing serious adverse events, which it didn’t (please see table 6, line 238). With regards to the unsolicited and unexpected adverse events, due to their sparsity we do not see any relationship with previous comorbidities but suspect some may be related to their underlying cancer diagnosis or treatment. For example, worsening existing tumour pain which may represent tumour inflammation from immune reconstitution or hypocortisolism in a patient on immunotherapy (please see table 4, line 174).

With regards to the second comment, we have not done a long term follow-up yet for these patients as we await the second vaccine dose but that will be the aim for the future of our safety monitoring study. Please see our reply to Reviewer 1 and 2. We have added the following sentence to the latest paragraph in the discussion to address this: “Another limitation to our study is that the majority of our patients have not yet received a second vaccine dose due to current vaccination strategies to increase population uptake of the first dose.” (line 382-384).

We have included human ethics information at the start and end of the manuscript (lines 84 and 403). This study was approved under the Guy’s Cancer Cohort. As this was a retrospective study with no additional investigations or intervention from standard of care, no further ethical requirements were needed.

Unfortunately, due to the observational design no antibody titres were collected from vaccinated individuals in this study. In addition, antibody titres are not routinely collected as part of standard of care in our country and centre. Therefore, we do not know whether adverse events correlate with the antibody titres in these inviduals. We have now described this as a limitation and suggestion for future studies in the discussion section: “Unfortunately, due to the observational nature of our study we do not have the antibody titers of our vaccinated patients to do further regression analysis. This may be an important component to include in future studies.” (line 354-357).

Round 2

Reviewer 1 Report

I accept the authors' response to my comments.